# Corneal Epithelial Toxicity after Intravitreal Methotrexate Injection for Vitreoretinal Lymphoma: Clinical and In Vitro Studies

**DOI:** 10.3390/jcm9082672

**Published:** 2020-08-18

**Authors:** Yoon Jeong, Jin Suk Ryu, Un Chul Park, Joo Youn Oh

**Affiliations:** 1Department of Ophthalmology, Seoul National University College of Medicine, 103 Daehak-ro, Jongno-gu, Seoul 03080, Korea; pixieyoon@naver.com; 2Laboratory of Ocular Regenerative Medicine and Immunology, Biomedical Research Institute, Seoul National University Hospital, 101 Daehak-ro, Jongno-gu, Seoul 03080, Korea; enter2357@naver.com; 3Retinal Degeneration Research Laboratory, Biomedical Research Institute, Seoul National University Hospital, 101 Daehak-ro, Jongno-gu, Seoul 03080, Korea

**Keywords:** apoptosis, corneal epithelium, folic acid, folinic acid, methotrexate, vitreoretinal lymphoma

## Abstract

Methotrexate is widely used as an intraocular chemotherapy for vitreoretinal lymphoma (VRL). Although corneal toxicity has been reported in patients after intravitreal methotrexate injections, the incidence, outcome, and mechanism of the toxicity are unclear. Herein, we performed a clinical study to evaluate the incidence, predisposing factors, and treatment outcome of corneal epitheliopathy associated with intravitreal methotrexate injection. In addition, we directly investigated cytotoxic effects and mechanisms of methotrexate in cultures of human corneal epithelial cells (CECs). Medical chart reviews revealed that corneal epitheliopathy occurred in 15 eyes (22.7%, 12 patients) out of 66 eyes (45 patients) after intravitreal methotrexate injections for treatment of VRL. The use of topical anti-glaucoma medication was significantly associated with development of corneal epitheliopathy. The epitheliopathy resolved in all patients 2.4 months after onset. In culture, methotrexate decreased the survival of CECs by inducing apoptosis, increasing oxidative stress, suppressing proliferation, and upregulating inflammatory cytokines. The addition of folinic acid significantly protected the cells from the methotrexate-induced toxicity. Hence, our results suggest that care should be taken to minimize the contact of methotrexate with corneal epithelium during injection, and folic or folinic acid supplementation might be beneficial for preventing corneal complications in patients undergoing intravitreal methotrexate injections.

## 1. Introduction

Methotrexate (MTX) is a competitive inhibitor of dihydrofolate reductase (DHFR) and has an anti-neoplastic effect in multiple types of cancers by reducing intracellular folic acid levels and thereby disrupting DNA and RNA synthesis in cancer cells [1,2,3]. In ophthalmology, intravitreal injection of MTX is widely used as an intraocular chemotherapy for vitreoretinal lymphoma (VRL) as it has been shown highly effective in inducing complete remission of the tumor and avoiding systemic complications [4,5,6]. On the other hand, intravitreal MTX is associated with a wide range of ocular complications including cataract, neovascular glaucoma, sterile intraocular inflammatory reaction, and keratopathy [4,7,8]. In particular, corneal epitheliopathy is one of the most common and serious side effects observed after intravitreal MTX injection [4,7,8,9].

MTX-induced cytotoxicity and its mechanisms have been reported in a number of non-cancer cell types such as hepatocytes, alveolar epithelial cells, and intestinal cells [10,11,12]. Yet it is unclear whether MTX has a direct toxic effect on corneal epithelial cells (CECs). Moreover, few studies investigated clinical course, predisposing factors, or prognosis of MTX-associated corneal epitheliopathy. Hence, we herein performed a clinical study to evaluate the incidence, risk factors, and treatment outcome of corneal epitheliopathy associated with intravitreal MTX injection in patients with VRL. In parallel, we carried out an in vitro study to directly analyze the cytotoxic effects of MTX on human CECs and its mechanism of action.

## 2. Materials and Methods

This study was approved by the Institutional Review Board of Seoul National University Hospital (No. 1907-182-1051) and was conducted in accordance with the tenets of the Declaration of Helsinki.

### 2.1. Clinical Study

This was a retrospective chart review study. Inclusion criteria were patients who received their first intravitreal MTX injection for treatment of VRL from May 2010 to October 2019 in Seoul National University Hospital. Exclusion criteria included patients under the age of 18 years or if patients had a preexisting corneal disease or systemic disease associated with ocular surface damage (Sjögren’s syndrome, Stevens–Johnson syndrome, ocular graft-versus-host disease, etc.). Diagnosis of VRL was made based on the presence of malignant lymphoma cells from vitreous biopsy determined by an experienced pathologist. If vitreous biopsy was not diagnostic, supplemental tests such as IL-10 measurement and PCR analysis for monoclonality in the vitreous fluid were also taken into account when making a diagnosis of VRL [13,14]. An intravitreal injection of MTX was carried out following the standard protocol as previously reported [15]. Briefly, 400 μg MTX in 0.1 mL (Pfizer, West Ryde, Australia) was used for injection during the following three phases: (1) twice-weekly injection for one month of induction phase, (2) weekly injection for 1–2 months of consolidation phase, and (3) monthly injection for 9–12 months of maintenance phase. The number, interval, and duration of injections were adjusted at the discretion of a treating retinal specialist.

From the medical records and digital anterior segment photographs, data were collected as follows: demographic parameters (age and gender), systemic and ocular medical history, ocular findings (disease laterality, intraocular and retinal findings, corneal findings, and visual acuity), characteristics related to MTX injection (the number, interval, and duration of injections), concomitant use of topical medications affecting the ocular surface, treatment method for corneal epitheliopathy, and follow-up duration. Patients were divided into two groups based on development of corneal epitheliopathy, and the aforementioned factors were compared between eyes with and without corneal epitheliopathy.

### 2.2. Cell Culture

Human CECs were primarily cultured from the corneolimbal rim obtained from deceased human donors with informed consent as previously reported [16]. Briefly, epithelial cells were dissociated from the stroma by incubation of the corneolimbal tissue in 10 mg/mL dispase II (Cat# 12403300, Roche, Indianapolis, IN, USA) and DMEM/F12 (Welgene, Daegu, Korea) at 37 °C for 1 h followed by 0.2 mg/mL EDTA (Cat# 17-711, Lonza, Valais, Switzerland) at 37 °C for 5 min. Then, the epithelial cells were seeded on a mitomycin C (Sigma-Aldrich, St Louis, MO, USA)-pretreated 3T3 fibroblast feeder layer (NIH/3T3 cell line, ATCC, Manassas, VA, USA) and cultured for 2–3 weeks in supplemented hormonal epithelial media (SHEM) containing DMEM/F12 (Welgene, Daegu, Korea), 10% fetal bovine serum (Welgene), 0.5% penicillin-streptomycin (Lonza), 5 µg/mL insulin, 30 ng/mL cholera toxin, 10 ng/mL epidermal growth factor, 0.18 mM adenine, 4 mM L-glutamine, and 2 nM triiodo-L-thyronine (all supplements from Sigma-Aldrich). Passage 2 cells were treated with incremental doses of MTX (0, 10, 100, 300, and 1000 μM; Cat# M8407, Sigma-Aldrich) at 37 °C for 2 h or 24 h and subjected to analysis. MTX was dissolved in 1 M NH_4_OH. The test doses of MTX were determined to range from 10–1000 μM based on the following: (1) The intravitreal dose of MTX commonly used in clinics is 400 μg/0.1 mL which amounts to 8802 μM. (2) According to a previous pharmacokinetics study, MTX concentration in the vitreous cavity immediately after 800 μg/0.1 mL intravitreal injection was 1321 ± 185 μM [17]. (3) In a preliminary study, 10,000 μM MTX killed most human CECs, making further experiments impossible.

In some groups, different doses of folinic acid (FA) (0, 1000, 3000, 10,000, and 30,000 μM; Cat# 47612, Sigma-Aldrich) were added to cultures. FA was used to replenish folic acid in the MTX-treated cells because it is the reduced form of folic acid and circumvents the inhibition of DHFR. Both untreated CECs and vehicle (1 M NH_4_OH)-treated CECs were used as controls. The pH was maintained 7.4–7.6 in both SHEM-treated and vehicle-treated cell cultures.

### 2.3. Cell Viability and Proliferation Assays

The metabolic activity of CECs, indicative of cell viability, was assessed by a colorimetric assay that measures the reduction of tetrazolium salt (WST-8) by dehydrogenase activities in cells (Cell Counting Kit-8, Cat# CK04, Dojindo Molecular Technologies, Rockville, MD, USA). Cell proliferation was quantified by measurement of BrdU incorporation into cells using a colorimetric immunoassay (Cell Proliferation ELISA, BrdU, Cat# 11 647 229 001, Roche).

### 2.4. Apoptosis Assay and Oxidative Stress Measurement

For evaluation of cell apoptosis, CECs were stained with Annexin V (ANX) and propidium iodide (PI) using a FITC Annexin V Apoptosis Detection Kit (Cat# 556547, BD Pharmingen™, San Diego, CA, USA) and analyzed for apoptotic ANX^+^PI^+^ cells by flow cytometry (S1000EXi Flow Cytometer, Stratedigm, San Jose, CA, USA). For cellular and mitochondrial reactive oxygen species (ROS) measurement, CECs were stained with 5 μM CellROX dye (CellROX^®^ Deep Red Reagent, Cat# C10422, Thermo Fisher Scientific, Carlsbad, CA, USA) and 100 nM Mitotracker (MT) Green dye (MitoTracker™ Green FM, Cat# M7514, Thermo Fisher Scientific) and assayed for CellROX^+^MT^+^ cells by flow cytometry (S1000EXi Flow Cytometer). Flow cytometric results were analyzed using Flowjo software (Tree Star, Ashland, OR, USA).

### 2.5. Real-Time Reverse Transcription (RT)-PCR

The mRNA levels of TNF-α and IL-6 were evaluated by real-time RT-PCR. CECs were lysed in RNA isolation reagent (RNA Bee, Tel-Test Inc., Friendswood, TX, USA), and total RNA was extracted using RNeasy Mini kit (Qiagen, Valencia, CA, USA). First-strand cDNA was synthesized by reverse transcription (High Capacity RNA-to-cDNA Kit, Applied Biosystems, Carlsbad, CA, USA), and real-time amplification for each gene was performed using TaqMan^®^ Universal PCR Master Mix (Applied Biosystems) in an ABI 7500 Real-Time PCR System (Applied Biosystems). The PCR probe and primer sets were purchased from Applied Biosystems (TaqMan^®^ Gene Expression Assay), and GAPDH was used as the reference RNA. After normalization by GAPDH, the gene expression value in each sample relative to that in controls was calculated by the comparative ΔΔCt method.

### 2.6. Statistical Analysis

Statistical analysis was conducted by SPSS 20.0 Software (SPSS, Inc., Chicago, IL, USA) for clinical data and GraphPad Software (GraphPad Prism^®^, Inc., La Jolla, CA, USA) for in vitro data. The comparison between two independent groups was made by the Mann–Whitney *U* or two-tailed Student’s *t-*test for quantitative variables and by a Chi-square or Fisher’s exact test for qualitative variables. The comparison of means of more than two groups was performed by using one-way ANOVA and Tukey’s Honestly Significant Difference test. Visual acuity was converted to the logarithm of minimal angle of resolution (logMAR) scale for statistical analysis and compared in the same patient before and after treatment by the Wilcoxon signed rank test. Data are presented as mean ± SD. A *p* value < 0.05 was considered statistically significant.

## 3. Results

### 3.1. Demographics and Clinical Characteristics

In total, 66 eyes of 45 patients were included in the study (21 patients had binocular involvement of VRL). Demographics, systemic medical history, and clinical characteristics are presented in Table 1. The mean age at first injection of MTX was 60.4 ± 11.6 years (27–85 years). Among the total of 45 patients, male and female patients numbered 20 (44.4%) and 25 (55.6%), respectively. Among the total of 66 eyes, 34 occurred in the right eye and 32 in the left eye. Eight eyes (12.1%) were diagnosed with primary VRL affecting the eye without involvement of other organs, and 58 eyes (87.9%) were diagnosed concurrently with or secondary to central nervous system (CNS) lymphoma. The average number of MTX injection in a patient was 8.4 ± 5.7 (1–23). The median and mean follow-up durations after first MTX injection were 14.9 months and 19.1 months, respectively.

### 3.2. Incidence, Characteristics, and Risk Factors of MTX-Induced Corneal Epitheliopathy

Corneal epitheliopathy occurred in 15 (22.7%, 12 patients) of 66 eyes (45 patients). The average number of MTX injections before development of epitheliopathy was 5.7 ± 3.4 (1–12). The epitheliopathy developed at the mean 5.7 ± 4.7 weeks (3–123 days) after the first MTX injection and at the mean 11.3 ± 13.6 days (1–56 days) after the last injection. All of the corneal epitheliopathy manifested as superficial punctate epithelial erosions (*n* = 15, 100%), some of which were accompanied by filaments (*n* = 3, 20%), central epithelial defects (*n* = 3, 20%), and epithelial sheet-like opacity (*n* = 2, 13.3%). Representative photographs are shown in Figure 1 (Figure 1A,C,E,G).

For identification of predisposing factors associated with corneal epitheliopathy, demographic and clinical parameters were compared between eyes with corneal epitheliopathy and those without epitheliopathy (Table 1). Among the analyzed factors, the concomitant use of topical anti-glaucoma medication at time of MTX injection was more frequently found in eyes with corneal epitheliopathy, compared to those without epitheliopathy. Four out of 15 eyes (26.7%) in the epitheliopathy group used topical anti-glaucoma eye drops, while only 2 of 50 eyes (3.9%) did in the non-epitheliopathy group (*p* = 0.021). Other factors such as age, gender, disease laterality (right, left, or both), diagnosis (primary or secondary VRL), fundus findings, or systemic comorbidity (diabetes mellitus) did not show a significant correlation with development of corneal epitheliopathy. Of note, the number of MTX injections before epitheliopathy development (5.7 ± 3.4) was not different from the total number of MTX injections in the non-epitheliopathy group (7.4 ± 5.0). However, the total number of injections during the whole VRL treatment period was significantly higher in the epitheliopathy group (11.7 ± 6.7) than in the non-epitheliopathy group (7.4 ± 5.0, *p* = 0.011), suggesting that patients in the epitheliopathy group required more MTX injections for VRL treatment than those in the non-epitheliopathy group.

### 3.3. Treatment and Outcome of MTX-Induced Corneal Epitheliopathy

Treatment methods for corneal epitheliopathy were as follows: temporary cessation of MTX injection (*n* = 10, 66.7%), prophylactic use of topical antibiotics (*n* = 10, 66.7%), preservative-free artificial tears (*n* = 7, 46.7%), autologous serum eye drops (*n* = 7, 46.7%), topical corticosteroids (*n* = 6, 40%), therapeutic contact lens (*n* = 6, 40%), oral folic acid supplementation (*n* = 4, 26.7%), and topical anti-viral agents (*n* = 4, 26.7%).

Corneal epitheliopathy resolved in all 15 affected eyes at the median 1.9 months and at the mean 2.4 ± 1.6 months (20 days to 5.7 months) after treatment. Representative corneal photographs after treatment are shown in Figure 1 (Figure 1B,D,F,H). LogMAR visual acuities significantly improved from 1.29 ± 0.63 (3.00–0.22) to 0.69 ± 0.66 (2.00–0.1) after resolution (*p* = 0.006). None of patients had a recurrence of keratopathy.

### 3.4. MTX-Induced Cytotoxicity in CECs

In parallel with the clinical study, we examined whether MTX might exert direct toxicity in human CECs (Figure 2A). The measurement of metabolic activity showed that MTX suppressed the viability of CECs in time- and concentration-dependent manners (Figure 2B). The cell viability was reduced to 66.7% relative to vehicle-treated controls after 2 h of 300 μM MTX treatment and further decreased to 45.2% after 24 h. The reduction of cell viability was more marked with 1000 μM MTX; the viability was 32.3% and 19.4% after 2 h and 24 h of 1000 μM MTX treatment, respectively, compared to vehicle treatment. A similar observation was made with cell proliferation. The BrdU uptake assay indicated that MTX inhibited the proliferation of CECs in a concentration-dependent manner with 300 μM and 1000 μM MTX exhibiting significant inhibitory effects (Figure 2C). In addition, MTX induced the apoptosis of CECs. The frequency of ANX^+^PI^+^ cells, indicative of apoptotic cells, was significantly higher in CECs treated with 300 μM and 1000 μM MTX (48.1 ± 7.1% and 58.9 ± 4.6%) than in vehicle-treated cells (20.4 ± 4.4%) as assessed by flow cytometry (Figure 2D,E). Similarly, the frequency of CellROX^+^MT^+^ cells, reflecting mitochondrial ROS generation, was significantly higher in 300 μM and 1000 μM MTX-treated CECs (31.6 ± 4.5% and 41.5 ± 6.1%), compared with the vehicle-treated control (16.7 ± 3.2%) (Figure 2D,F).

Clearly, the data demonstrate that MTX has negative effects on the survival of human CECs through suppression of proliferation, induction of apoptosis, and generation of oxidative stress.

### 3.5. Protective Effect of FA on MTX-Induced Cytotoxicity in CECs

MTX depletes intracellular folic acid and thereby interferes with synthesis of purine and pyrimidine, which are important DNA precursors in tissues with a high cellular turnover rate [2]. In addition, it has been shown that folic acid suppresses MTX influx [11]. In line with these mechanistic data, there is a large body of clinical evidence that oral supplementation of folic acid or FA is beneficial for reducing MTX-induced adverse effects in patients with rheumatoid arthritis and cancer [18,19,20,21]. Based on this knowledge, we next tested whether FA replenishment might prevent the MTX-induced cytotoxicity in CECs (Figure 3A). Results revealed that an addition of 1000 μM or 3000 μM FA to the cultures effectively preserved cell viability and proliferation in 300 μM MTX-treated CECs while inhibiting apoptosis and mitochondrial ROS generation (Figure 3B–D). Likewise, 10,000 μM and 30,000 μM FA significantly abrogated the MTX-induced toxicity in 1000 μM MTX-treated CECs (Figure 3B–D).

Hence, the results suggest that FA protects human CECs against the MTX-mediated toxicity.

### 3.6. Effect of MTX and FA on Inflammatory Cytokine Expression in CECs

We additionally tested the effects of MTX on the expression of pro-inflammatory cytokines in human CECs (Figure 3E). Real-time RT-PCR analysis showed that the mRNA levels of IL-6 and TNF-α were significantly higher in MTX-treated cells than in vehicle-treated cells, an indication that MTX elicited the expression of pro-inflammatory cytokines in CECs. The addition of FA significantly attenuated the effect of MTX on the induction of inflammatory cytokines in the cells.

## 4. Discussion

Our data demonstrate that 22.7% of eyes developed corneal epitheliopathy after a mean 5.7 injections (1–12) of intravitreal MTX. Thus far, our study, involving 66 eyes of 45 patients, is one of the largest case series in the published literature on the intravitreal MTX-associated corneal complications. A similar study by Smith et al. observed corneal epitheliopathy in 15 of the 26 MTX-treated eyes (58%) [7]. Another study by Frenkel et al. showed that some form of keratopathy, ranging from mild dryness to severe epitheliopathy, appeared in all VRL patients (44 eyes of 26 patients) usually after the third MTX injection [4]. As it has been reported that clinical remission of VRL was achieved after an average 6.4–8.5 intravitreal MTX injections and a maximum of 16 injections per eye [4,7], corneal epitheliopathy would be an inevitable complication following intravitreal MTX injection that can preclude the proper injection schedule for VRL remission. It would be interesting to further investigate why corneal epitheliopathy develops after several injections of MTX. It might be attributed to the cumulative effects of MTX on corneal epithelial cells, or hypersensitivity reaction might be involved in the MTX toxicity.

Regarding the mechanism responsible for the MTX-induced corneal epitheliopathy, our study reveals that MTX directly impairs the survival of mitotically active limbal epithelial progenitors and CECs through repression of proliferation and induction of apoptosis and oxidative stress, which are consistent with previous observations of MTX toxicity in hepatocytes, lung epithelial cells, and intestinal epithelial cells [10,11,12,22]. Since the toxic effects of MTX on corneal epithelial cells were dependent on the MTX concentration in our study and the anti-neoplastic effects of MTX are known to be dose-dependent, it is possible that the epitheliopathy might develop in patients in an MTX dose-dependent manner. Thus, further study would be interesting to investigate the relationship between MTX doses used and the severity of epitheliopathy. Furthermore, considering the detrimental impact of MTX on limbal and corneal epithelium, it would be important to avoid spillage of MTX onto the limbal and corneal surface and to prevent the subconjunctival leakage of MTX during intravitreal injection.

In addition, we found that MTX stimulated the production of pro-inflammatory cytokines from CECs, suggesting inflammatory damage to the ocular surface. Therefore, it is possible that MTX impairs the cell viability through induction of an inflammatory response in addition to the direct cytotoxicity.

Intriguingly, these toxic mechanisms of MTX in CECs such as apoptosis induction and inflammatory activation are similar to what we previously observed with benzalkonium chloride-containing anti-glaucoma medication [16]. In the present study, the use of topical anti-glaucoma medication at time of MTX injection was significantly associated with development of corneal epitheliopathy after intravitreal MTX injection. These findings collectively suggest that patients using the anti-glaucoma eyedrops should be more closely monitored for corneal epitheliopathy after MTX injection.

MTX inhibits DHFR, a key enzyme in folic acid metabolism, thereby depleting the intracellular pool of reduced folates and producing a state of folate deficiency [2]. Folic acid is essential for the synthesis of purines and pyrimidines, essential DNA precursors. Oral supplementation of folic acid is well-known to ameliorate the toxicity of MTX in patients with psoriasis, inflammatory arthritis, or cancer [18,19,20,21]. Indeed, our in vitro data reveal that FA protects human CECs from MTX-induced cytotoxicity. In addition, our clinical data showed that folic acid supplementation resulted in the resolution of corneal epitheliopathy in all five patients who had been treated with oral folic acid. In accordance with these data, there have been several case reports showing the efficacy of topical or oral folic acid in the management of intravitreal MTX-induced corneal epitheliopathy [7,8,9]. Further studies will be necessary to determine the optimal dose and administration route of folic acid and FA for effective prevention of the MTX-induced ocular side effects without compromising the therapeutic efficacy of MTX in the remission of VRL. Topical administration of folic acid or FA can be one of the options.

In summary, corneal epitheliopathy developed in 22.7% of VRL patients after intravitreal MTX injection, and 66.7% of the corneal epitheliopathy was severe enough to require the cessation of MTX injection. Considering direct cytotoxicity of MTX in human CECs, we propose that caution should be exercised to minimize the contact of MTX with corneal and limbal epithelial cells during or immediately after intravitreal injection. Additionally, folic acid or FA supplementation would be beneficial for preventing and treating the toxicity.

## Figures and Tables

**Figure 1 jcm-09-02672-f001:**
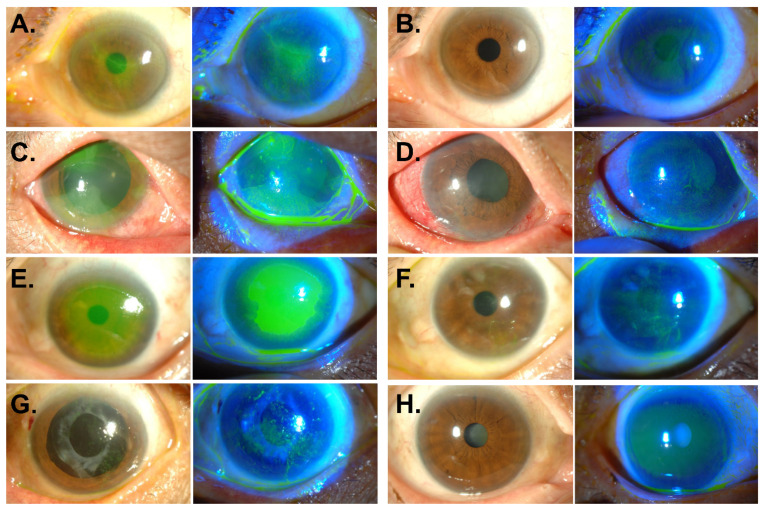
Anterior segment photography of corneal epitheliopathy after intravitreal methotrexate injection. (**A**,**B**) Left eye of 58-year-old female patient. Multiple superficial punctate epithelial erosions were observed 2 weeks after the tenth MTX injection (**A**). Three weeks after treatment with oral folic acid supplementation and therapeutic contact lens, corneal epitheliopathy completely resolved (**B**). (**C**,**D**) Right eye of 49-year-old male patient. Diffuse punctate epithelial erosions, grey-white plaque-like epithelial opacity, and conjunctival injection developed 3 days after the 8th MTX injection (**C**). Five months after treatment with therapeutic contact lens, autologous serum eyedrops, and topical corticosteroids, corneal opacity subsided (**D**). (**E**,**F**) Left eye of 64-year-old female patient. Large epithelial defect with clear-cut margin was present at the center of the cornea 7 days after the 3rd MTX injection (**E**). Two months after treatment with therapeutic contact lens, artificial tears, autologous serum eyedrops, and topical corticosteroids, the epithelium was healed (**F**). (**G**,**H**) Left eye of 76-year-old female patient. Multiple fine filaments with superficial punctate epithelial erosions appeared 14 days after the 4th MTX injection (**G**). The epitheliopathy was completely treated after 1 month of treatment with autologous serum eyedrops and oral folic acid (**H**).

**Figure 2 jcm-09-02672-f002:**
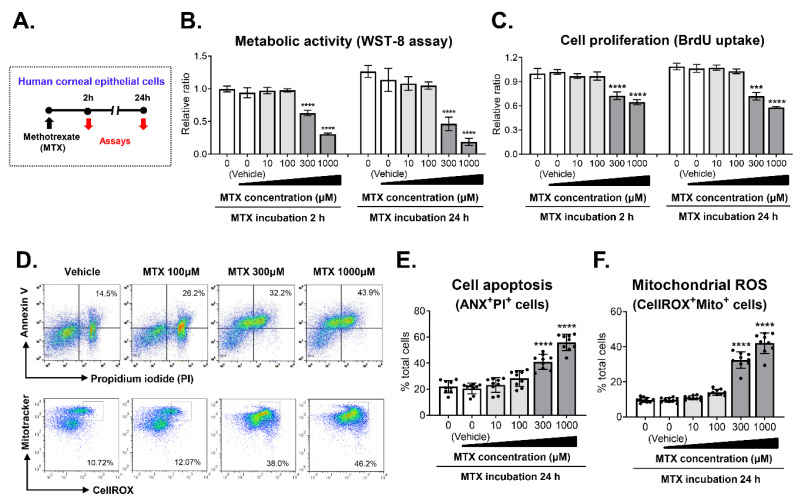
Toxic effects of methotrexate on human corneal epithelial cells. (**A**) Experimental scheme. Primary cultures of human corneal epithelial cells were treated with various concentrations of methotrexate (MTX) and subjected to assays 2 h and 24 h later. (**B**) Metabolic activity of cells as determined by WST-8 assay. (**C**) Cell proliferation as measured by BrdU uptake. (**D**–**F**) Representative cytograms and quantitative results for cell apoptosis and mitochondrial reactive oxygen species (ROS) as analyzed by flow cytometry. Data are presented as mean ± SD and from 3 independent sets of experiments (*n* = 3–5 per group per set). A dot depicts data from one biological sample. *** *p* < 0.001, **** *p* < 0.0001 by one-way ANOVA and Tukey’s multiple-comparison test.

**Figure 3 jcm-09-02672-f003:**
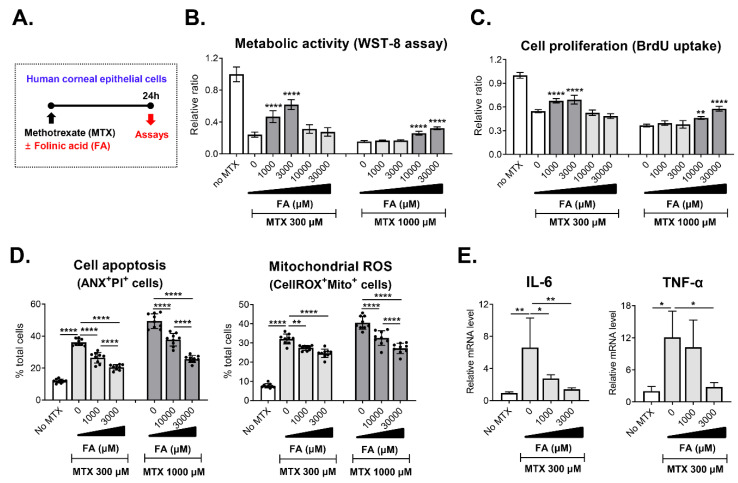
Protective effects of folinic acid on methotrexate-induced toxicity in corneal epithelial cells. (**A**) Experimental scheme. Primary cultures of human corneal epithelial cells were treated with 300 μM or 1000 μM methotrexate (MTX) in the absence or presence of various concentrations of folinic acid (FA) and assayed 24 h later. (**B**) WST-8 assay for measurement of cell metabolic activity. (**C**) BrdU uptake assay for assessment of cell proliferation. (**D**) Quantitative flow cytometry results for cell apoptosis and mitochondrial reactive oxygen species (ROS) production. (**E**) Real-time RT-PCR analysis for IL-6 and TNF-α in human corneal epithelial cells. Shown are data scaled to MTX- and FA-untreated cells. Data are presented as mean ± SD and from 3 independent sets of experiments (*n* = 3 per group per set). A dot depicts data from one biological sample. * *p* < 0.05, ** *p* < 0.01, **** *p* < 0.0001 by one-way ANOVA and Tukey’s multiple-comparison test.

**Table 1 jcm-09-02672-t001:** Clinical characteristics and risk factor analysis for corneal epitheliopathy in vitreoretinal lymphoma (VRL) patients after intravitreal methotrexate (MTX) injection.

Clinical Characteristics	Total	No Keratopathy	Keratopathy	*p* Value
No of eyes (patients)	66 (45)	51 (33)	15 (12)	
Age at first injection (years, range)	60.4 ± 11.6 (27–85)	61.4 ± 12.4 (27–85)	56.9 ± 7.8 (48–76)	0.100 *
Gender (male/female)	20:25	15:18	5:7	0.821 ^†^
Laterality (right/left)	34:32	27:24	7:8	0.669 ^†^
Total No of MTX injections (range)	8.4 ± 5.7 (1–23)	7.4 ± 5.0 (1–23)	11.7 ± 6.7 (1–22)	0.011 ^‡^
Injection No before keratopathy (range)			5.7 ± 3.4 (1–12)	
Fundus findings				
Vitreous haze	65 (98.5%)	50 (98.0%)	15 (100.0%)	1.000 ^§^
Subretinal/retinal infiltration	26 (39.4%)	19 (37.3%)	7 (46.7%)	0.512 ^†^
Retinal vasculitis	7 (10.6%)	5 (9.8%)	2 (13.3%)	0.653 ^§^
Exudative detachment	1 (1.5%)	0 (0.0%)	1 (6.7%)	0.227 ^§^
Indication for intravitreal MTX				
Primary VRL	8 (12.1%)	5 (9.8%)	3 (20.0%)	0.368 ^§^
VRL with CNS lymphoma	58 (87.9%)	46 (90.2%)	12 (80.0%)	0.368 ^§^
Anti-glaucoma eye drops	6 (9.1%)	2 (3.9%)	4 (26.7%)	0.021 ^§^
Diabetes mellitus	6 (9.1%)	3 (5.9%)	3 (20.0%)	0.125 ^§^
Duration from first injection to last follow-up (months, range)	19.1 ± 16.9 (0.1–79.3)	16.1 ± 14.8 (0.1–61.4)	29.2 ± 19.9 (6.3–79.3)	0.006 ^‡^

* Student’s *t*-test; ^†^ Chi-square test; ^‡^ Mann–Whitney *U* test; ^§^ Fisher’s exact test.

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
