# Peer review of "Corneal Epithelial Toxicity after Intravitreal Methotrexate Injection for Vitreoretinal Lymphoma: Clinical and In Vitro Studies"

_jcm, 2020, doi:10.3390/jcm9082672_

Round 1

Reviewer 1 Report

Line 110:  at this location or elsewhere in the paper, it would be helpful to contextualize these MTX concentrations. How do these equate to what might be seen in vivo, in intraocular space following injection, or in the bloodstream following intraocular injection?

Line 326:  how does the time course of the complications seen in patients compare with the findings in vitro? Is it reasonable to conclude that the same process is underway? Why is it not until the 5th or even the 10thinjection that the complication occurs?

Line 331:  is that speculative? Do we know that the route of access would be direct exposure to the limbus, or that these recommended steps would make a difference at all?

Line 340:  implies that the mechanism of glaucoma medication exacerbating the epithelial toxicity is via stimulation of inflammation. But could it not also be a direct chemical effect from the benzalkonium chloride or other components of the eyedrops?

Line 365:  same comment as for 331. Is brief contact with the limbus equivalent to the 2 hour incubation used in the in vitro study?

Author Response

Response to Reviewer 1 Comments

We thank the reviewer for time and excellent points which helped us to improve the paper.

Point 1: Line 110:  at this location or elsewhere in the paper, it would be helpful to contextualize these MTX concentrations. How do these equate to what might be seen in vivo, in intraocular space following injection, or in the bloodstream following intraocular injection? 

Response 1: We determined the range of MTX concentrations to be tested in vitro based on the following: 1) the intravitreal dose for MTX injection is 400 μg/0.1 mL which amounts to 8802 μM.  2) According to the pharmacokinetics study (Indian J Ophthalmol. 2011 May-Jun; 59(3): 197–200), the intravitreal MTX concentration right after MTX 800 μg/0.1 mL injection is 1321 μM, approximately one tenth of injected concentration. Therefore, we tested MTX up to 1000 μM in our in vitro studies (In a preliminary study, we also tested 10000 μM which killed most of corneal epithelial cells, making further experiments impossible). As advised, we included these points in the revised manuscript (Line 115-121).

Point 2: Line 326:  how does the time course of the complications seen in patients compare with the findings in vitro? Is it reasonable to conclude that the same process is underway? Why is it not until the 5th or even the 10th injection that the complication occurs?

Response 2: This was tricky for us to explain or speculate. But according to other literature and our own findings, it is a common finding that corneal epitheliopathy occurs after several injections (3 – 5 injections on average).  This might be due to the cumulative effects of MTX (MTX injected twice a week for the first month and once a week for the next 1-2 months. Although the half-life of MTX is 10 hours, other studies showed that the effective intravitreal concentration of MTX for eradication of lymphoma cells is maintained for 5 days after intravitreal injection). Or it is possible that some allergic or hypersensitivity reaction might be involved in the MTX toxicity to the corneal epithelium (similar to another anticancer drug, mitomycin C). Although our present study is limited to testing direct toxicity of MTX in corneal epithelial cells, further studies assessing pharmacokinetics of MTX in the ocular surface and evaluating the effects of MTX on host immune cells would be necessary to understand the complete action mechanism of MTX in the induction of corneal toxicity. We added this speculation in the discussion of the revised manuscript (Line 295-298). Thank you for allowing us to speculate on this important point.

Point 3: Line 331:  is that speculative? Do we know that the route of access would be direct exposure to the limbus, or that these recommended steps would make a difference at all?

Response 3: That is a speculative description and there is no proof that this technique (making anterior chamber paracentesis after injection to minimize the subconjunctival leakage of a drug) makes a difference.  Hence, as the reviewer wisely pointed out, we deleted the sentence and stated instead that it is important to minimize the direct exposure of MTX to the corneal and limbal epithelium by taking care not to overspill MTX to the ocular surface and preventing the subconjunctival leakage (Line 309-312).  

Point 4: Line 340:  implies that the mechanism of glaucoma medication exacerbating the epithelial toxicity is via stimulation of inflammation. But could it not also be a direct chemical effect from the benzalkonium chloride or other components of the eyedrops?

Response 4: Sorry to cause confusion. We implied that the mechanisms of MTX toxicity are similar to BAK-containing glaucoma medications because both induced apoptosis in corneal epithelial cells and provoked inflammatory activation as our group observed with MTX in this study and with BAK in the previous study (Ref 16). We revised the sentence to correct the misunderstanding (Line 317-319).

Point 5: Line 365:  same comment as for 331. Is brief contact with the limbus equivalent to the 2 hour incubation used in the in vitro study?

Response 5: It is not a routine practice to wash out the ocular surface after intravitreal MTX injection in order to remove a drug that might have been spilled over to the ocular surface during the injection. Also, it is possible that MTX can be leaked into the subconjunctival space and in contact with the limbus. Hence, it is difficult to decide that MTX is in just a brief contact with the limbus (shorter than 2 hours) during and after MTX injection. Moreover, the half-life of MTX is 3 to 10 hours. We can do more experiments to see whether briefer exposure of MTX to corneal epithelial cells can still induce apoptosis in the cells. But at any rate, it is difficult to correlate the in vitro exposure time with the in vivo exposure time unless we perform the pharmacokinetics study of MTX in the ocular surface after intravitreal injection, which is beyond the subject of the current study. We appreciate the reviewer for bringing up this important issue and giving us the future subject of investigation.

Reviewer 2 Report

The authors present a well written and complete manuscript examining the negative effects of intravitreal methotrexate on the cornea, in particular, the epitheliopathy. Concurrently, there is an elegant experiment on the cytotoxic effects and mechanisms of methotrexate toxicity in cultures of human corneal epithelial cells. This is not entirely novel information, yet it does add some to the literature.

Comments:

  1. It appears that the group with keratopathy had a statically significantly higher number of injections. This point could be emphasized some in the discussion and conclusions that the toxicity may be dose related.

  1. Related to comment #1, it is suggested that the those using concurrent topical glaucoma medications had a statistically significant risk for having keratopathy . This might be confounded, as the group with keratopathy had a higher number of injections. It cannot be concluded that the glaucoma drops had a relation to the keratopathy without controlling for the number of methotrexate injections.

  1. It would be interesting in a future study to assess this prospectively, as well as to check endothelial toxicity with confocal microscopy.

Author Response

We thank the reviewer for time and excellent points which helped us to improve the paper.

Point 1: It appears that the group with keratopathy had a statically significantly higher number of injections. This point could be emphasized some in the discussion and conclusions that the toxicity may be dose related. 

Response 1: Sorry to make confusion. The number of MTX injection in the epitheliopathy group was 5.7 ± 3.4 before epitheliopathy development and 11.7 ± 6.7 in total (during the whole lymphoma treatment course including injections both before and after epitheliopathy development). The number of MTX injection in the non-epitheliopathy group was 7.4 ± 5.0 in total. The number of MTX injection before epitheliopathy development (5.7 ± 3.4) was not different from the total number of MTX injection in the non-epitheliopathy group (7.4 ± 5.0), whereas the total number of injection for the whole VRL treatment course was significantly higher in the epitheliopathy group (11.7 ± 6.7) than in the non-epitheliopathy group (7.4 ± 5.0, p = 0.011). We clarified these points in the revised manuscript (Line 208-213). Also, we emphasized the possibility of MTX dose-dependency of corneal toxicity in the discussion (Line 296-300).

Point 2: Related to comment #1, it is suggested that those using concurrent topical glaucoma medications had a statistically significant risk for having keratopathy. This might be confounded, as the group with keratopathy had a higher number of injections. It cannot be concluded that the glaucoma drops had a relation to the keratopathy without controlling for the number of methotrexate injections.

Response 2: As stated in Response 1, the number of MTX injection before epitheliopathy development in the epitheliopathy group (5.7 ± 3.4) was not different from the non-epitheliopathy group (7.4 ± 5.0), meaning that the group with keratopathy received the mean 5.4 injections before developing epitheliopathy, while the group without keratopathy received the mean 7.4 injections, yet not developing epitheliopathy.  To reduce confusion and make clarification, we restated these points in the revised manuscript (Line 208-213).

Point 3: It would be interesting in a future study to assess this prospectively, as well as to check endothelial toxicity with confocal microscopy.

Response 3: We agree with the reviewer that it is necessary to perform the prospective study with keeping the findings from the retrospective studies in mind and added this aspect in the discussion (Line 298-300). With regard to the corneal endothelial toxicity, it is possible that MTX is toxic to corneal endothelial cells as the reviewer wisely pointed out, although we could not find cases indicating the direct endothelial toxicity after MTX injection in the literature and in our series of patients.  
